# Changes in Patient Discourse: A Qualitative Study Based on the Treatment Experience of Chinese Patients with Somatization Symptoms

**DOI:** 10.3390/healthcare11212811

**Published:** 2023-10-24

**Authors:** Wenting Shu, Xiquan Ma, Xudong Zhao

**Affiliations:** 1School of Humanities, Tongji University, Shanghai 200092, China; tracy_shu1119@126.com; 2Department of Developmental and Behavioral Pediatrics, Shanghai Children’s Medical Center, Shanghai Jiao Tong University School of Medicine, Shanghai 200127, China; mxq919@163.com; 3Clinical Research Center for Mental Disorders, Shanghai Pudong New Area Mental Health Center, School of Medicine, Tongji University, Shanghai 200124, China; 4Department of Psychosomatic Medicine, Shanghai East Hospital, School of Medicine, Tongji University, Shanghai 200120, China

**Keywords:** somatization, treatment experience, discourse, interview, interpretative phenomenological analysis, qualitative study

## Abstract

This qualitative study examines the characteristics exhibited by Chinese patients with somatization symptoms during their treatment process, focusing on changes in illness interpretation and language use. A semi-structured in-depth interview was conducted with 10 patients receiving treatment in a clinical psychology department of a general hospital who reported somatic symptoms as their main complaint. The interview data were recorded and transcribed, and analyzed using interpretive phenomenological analysis. Two core themes emerged from the analysis: avoidance at the utterance level; and at the semantic level, power and contestation. Patients with somatization symptoms exhibit avoidance behaviors, and their experience of illness and the therapeutic process impact their discourse. Professionals should pay attention to patients’ own interpretations, cultural background and acceptance of the illness.

## 1. Introduction

In the 1960s, the Vienna School philosopher Gustav Bergmann proposed a shift in modern Western philosophy towards the philosophy of language, known as the ‘linguistic turn’. This shift has had a significant impact on research fields such as psychology and social sciences [1]. Following this shift, there has been increased research interest in narrative medicine and medical humanities, which focuses on patients’ experiences of illness and their individualized expression of subjectivity in the medical field [2].

In the field of psychiatry, somatization is defined as the presence of somatic symptoms that cannot be explained by organic causes [3]. Patients with somatic symptoms often experience denial of biological illness by medical doctors, and the psychiatrist’s explanation of somatization does not always align with the patient’s understanding and expectations. Consequently, patients may feel lost in psychiatric explanations, leading to multiple “medication-discontinuation-medication” sessions, confusion, and difficulty accepting the psychological explanation for their discomfort [4]. Unfortunately, the subjective experience of patients with physical symptoms is often overlooked, and their language is suppressed or submerged. Studies indicate that patients with somatic disorders perceive most of the doctor’s explanations as a denial of the reality of the symptoms, with only a small proportion of explanations being experienced as acceptance, leading to patient satisfaction [5].

There is also a low consistency between doctor and patient attribution of illness, with some patients rejecting the attribution of mental illness and preferring to attribute their symptoms to unknown diseases, accidents, excessive physical exertion, stress, and physical distress that lead to depression [6,7]. Turkish immigrant women with somatization symptoms reject attributions of mental illness, but prefer to attribute them to unknown diseases, accidents, excessive physical exertion, stress, and physical distress which lead to depression. Their attributions are influenced by Turkish folk interpretations and religious factors. They feel misunderstood in psychiatric treatment [8]. A study of male patients taking antidepressants found that taking the medication made them feel out of control. While medication helps male patients relieve emotional intensity and restore function, the men lose emotional vitality and sexuality. Patients are often in conflict with doctors, resulting in tense conversations on making autonomous decisions [9]. This suggests that there is a contradiction and disconnect between the discourse of psychiatry and the subjective feelings of patients.

Patients’ explanations of illness affect their choice of treatment. Patients with somatization symptoms interpret their illness in terms of age, gender, the stigma of mental illness, and pre-morbid social connections [10]. In addition, patients’ attributions also influence the illness belief and attitude, treatment-seeking preference, and treatment adherence [11]. For instance, Turkish female patients with somatization tend to prefer traditional therapy, hot spring healing, and the method of using amulets based on their religion [8]. Chinese patients with somatization have the option of traditional Chinese medicine, acupuncture, and qigong. Psychiatric treatment is the last hope [4]. Patients may feel unsure about whom to trust and may feel compelled to trust their doctors [8].

Patients with somatization symptoms present with physical symptoms as the main complaint. It is a physical and mental expression of “the lived body” [12]. In the context of Chinese culture, the body itself is endowed with complex meanings. The world as perceived by individuals is mediated by the body. Bodily experience is the main way for individuals to understand and communicate with the world. In contrast to the Western tradition of the dichotomy between the body and the mind, Chinese culture has been characterized by the mutual representation of the two from the beginning of its existence [13]. Western scholars regard the somatization of Chinese people as depression in a cultural context. This statement is defined by Western thinking, without considering the significance of the role of culture. There is the potential to broaden the diagnosis of depression [14,15]. In recent years, it has been proposed that applying the culture–mind–brain perspective to the study of Chinese somatization, which opens up new possibilities for the study of long-standing problems in cultural psychiatry. It has the potential to break down the false dichotomy between personal experience and overt expression of symptoms [16].

With the development of narrative medicine and psychotherapy, more research is focusing on the inner experience of patients, who have the same need to express themselves. It has been argued that psychoanalysis, the originator of psychotherapy, is the reproduction of a certain narrative tradition. This kind of talk therapy also plays an important role in the field of psychiatry today. There has been a long tradition of interpretation or narrative in history [17]. In studies of language used to describe ‘worsening asthma’, there are important differences in the language used by patients and doctors, and the choice of language vocabulary by patients is influenced by social context. This exacerbates the variability in doctor–patient communication and reduces the use of terminology [18].

In this context, as Chinese patients with somatization symptoms enter the medical field, their internal experiences, verbal expression, and etiological explanations unfold through their language narratives and expressions. Hence, the objective of this study is to interview patients with somatization symptoms regarding their experiences during illness and treatment processes to gain an understanding of the various dimensions of their language.

## 2. Materials and Methods

### 2.1. Study Design and Methodology

Interpretative phenomenological analysis (IPA) is performed in this study. IPA is an approach that originated in the field of health psychology and is dedicated to the study of how people understand their lived experiences [19]. The method advocates the use of purposive sampling with small homogeneous samples for in-depth and detailed analysis of individual cases so as to better present the individual’s understanding of their own experiences.

### 2.2. Participants

Patients with somatization symptoms were recruited from the clinical psychology department of a general hospital in Shanghai using purposive sampling. Psychiatry is rarely offered in Chinese general hospitals but clinical psychology, instead, in order to reduce people’s fear of mental illness. The clinical psychology department is concerned with psychosomatic medicine, which is the equivalent of psychosomatic medicine in the German medical system. Psychiatrists helped to ask any patients qualified under the inclusion criteria about their willingness to participate in the study interview. The inclusion criteria were as follows: (1) aged 18–60 years; (2) no organic problems have been detected by physicians; (3) patients with a diagnosis of “somatic symptom disorder, somatoform disorder, depressive disorder, panic disorder, anxiety disorder” as assessed by psychiatrists; and (4) patients with good cognitive and language skills. The following exclusion criteria were applied: patients with organic mental illness, substance dependence, schizophrenia, bipolar disorder, and eating disorder. The recruitment was from March, 2021 to August, 2021. No distinction was made as to whether participants were first-time visitors.

The sample size of the study was determined In accordance with the principle of “data saturation”, the state when the data appear to repeat and no new themes are presented during the data analysis. A total of 10 patients were eventually enrolled in the study. The information of patients is shown in Table 1. In order to protect the privacy of the patients, Amy, Bill, …, and John were used to refer to the interviewees when registering and quoting the interview scripts.

### 2.3. Data Collection

Semi-structured in-depth interviews were used. Each interview lasted 60–90 min, and each participant was interviewed 1–2 times. The number of interviews could be extended for patients who were interested in participating in depth. The interview began with the researcher introducing herself, asking for consent and establishing a trusting relationship before entering the formal interview. The interviews were audio-recorded with the consent of the participants and transcribed verbatim within 24 h of the interview. The second interview was conducted in a timely manner, an average of 15 days after the initial interview (range 10–21 days). Before the second interview, the information from the previous interview was reviewed, the key interview content was identified, and the interview outline was revised in time.

The outline of the interviews was as follows:*When did the physical symptoms first appear? Please describe the symptoms, the situation in which they appeared, the internal experience, feelings, emotions, thoughts and duration of symptoms, and the subsequent condition.**What treatments have been carried out during this period? Which hospitals and departments did you go to? What tests were performed? How long did the treatment take? Which treatments made you feel better? When did you consider coming to the clinical psychology department for treatment? What treatments were carried out in the clinical psychology department?**How did you perceive your symptoms in the beginning? How did you explain these symptoms? Did your interpretation change during the treatment? Do you think these symptoms are related to your mental status? If there is a relationship, what kind of relationship is it? If not, how do you understand it when you are being treated in a clinical psychology department? Where did you obtain these explanations?**During the consultation process, did the doctor explain your illness to you? What did you think of the doctor’s explanation? How did other people around you understand your disease? How do you feel about their understanding?**How have these symptoms affected your life? Have your friends and family treated you differently? What was it like before, and what is it like now? How do you feel about this? Do you feel that your relationship with them has changed compared to before?**Open-ended questions. Can you tell me a little bit about yourself? For example, your growth experience, things that have left a deep impression on you, your friends and family, the learning/career development process, gains, and setbacks.*

### 2.4. Data Analysis

According to the guidelines of Smith et al. [19], the IPA analysis starts with a single case, and completes a detailed analysis of the first text before starting to work on the next text. The specific analysis steps were as follows: (1) repeatedly read the transcript; (2) preliminary notes and comments; (3) propose the themes; (4) find the relationship between themes; (5) start the next case analysis; and (6) find inter-case thematic patterns.

### 2.5. Quality Control

(1) The main investigator of this study has systematically studied and trained in qualitative research and has rich experience in clinical psychological work. (2) The research supervisor of this study has extensive experience in qualitative research and is a clinical mental health practitioner. During the research process, the researcher and research supervisor discussed and analyzed the data numerous times and constantly compared and calibrated the data analysis results to ensure the accuracy of the analysis results.

## 3. Results

Respondents with somatization symptoms showed different levels of language deficits. Based on the IPA guidelines, the interview transcripts were analyzed, and two themes were identified. The results of this study including themes, sub-themes, sub-topics are shown in Table 2.

### 3.1. Utterance Level: Avoidance

At the level of utterance, all 10 respondents showed varying degrees of avoidance of language. In terms of verbal expressions, the respondents did not speak or avoided language in certain contexts. The 10 interviewees all began to seek medical treatment for physical discomfort, and they all received psychiatric treatment. During the interviews, all 10 interviewees expressed physical discomfort, the process of developing symptoms, and the experience of seeking medical treatment. This theme is divided into two sub-themes: behavioral avoidance and a tendency to avoid emotional language.

#### 3.1.1. Behavioral Avoidance

At total of eight interviewees showed behavioral avoidance, that is, avoidance, falling silent and not talking anymore. The specific performance was that they did not respond during the interview, avoided talking about specific topics, and avoided their discomfort with specific people.

Ivor, who was just 18 years old at the time of the interview and was about to enter university after graduating from high school, sought medical treatment because of persistent discomfort of “wrapping” in the back and snapping pain in the lower limbs. During the interview, Ivor’s mother was also present. In the interview, Ivor had a hard time answering questions about how he felt and how to understand. “I don’t know” and “I can’t explain clearly” were his responses. Instead, Ivor’s mother was able to talk about the process of his symptoms, the journey of seeking treatment, understanding his symptoms as related to high school stress, believing that Ivor was sensitive and nervous, and suggesting that he could relax. In response, when the researcher asked Ivor what he understood about his mother’s story, Ivor answered “Maybe”, “I don’t know”, “I don’t know”, and “I can’t do it.“ It seems that Ivor withdrew the language in his behavior, and it was difficult to directly understand his inner real thoughts and feelings. All he could express was simple physical discomfort, as well as a negative and difficult attitude.

Two of the interviewees avoided talking about certain topics during the interview. Bill and Emily were reluctant to talk about unpleasant topics, believing that “forgetting” these unpleasant topics was their usual way of dealing with them. Bill considered himself to be selective in his memory, speaking only about good experiences and indicating that he did not want to discuss relationship problems. During the interview, Emily was unwilling to mention her marital status. She thought that the love relationship was not important or pleasant, and she had forgotten about it. She responded that her mind was more on her parents and children.

Seven of the interviewees avoided talking about their discomfort with specific people, often family members. Interviewees could not be supported or understood by them when talking about their discomfort with them.

Three of the interviewees felt frustrated by their families’ inability to respond appropriately to them or give them support. Amy is a female high school graduate. She recounted how she had sought help from her parents after being verbally abused by her friends at school. The parents used a simple and brutal approach that did not help her reduce her stress but added to her difficulties. She felt frustrated with her family and friendships and could no longer trust and be close to others so she chose to avoid opening up her inner feelings. Cherry and Hebe both felt that their mothers could not understand or accept that they were sick. Cherry’s mother had a hard time because of her illness, which in turn, increased Cherry’s burden. Hebe’s mother denied her pain, believing that Hebe faked it. As a result, Cherry and Hebe avoided telling their mothers about their pain.

The other four interviewees were in the role of caregivers in their lives, taking care of their family’s emotions while avoiding their own pain and not wanting to add to the burden of others. Emily is a divorced woman in her forties with a child in high school who is very rebellious and often loses his temper at home. Emily had to bear her son’s emotions and hold her own. John’s father suffered from stomach cancer, which brought great pressure on John. He had to take care of his father, and he had to bear his discomfort silently. Bill stated in the interview, “*A man should be a tree, not a grass. To be a tree can shelter people from the wind and rain. The tree can hold up the sky*”. It was hard for Bill to reveal his difficulties. Daniel is also a person who took care of his family in his life.

In summary, 8 out of 10 respondents resorted to verbal avoidance when confronted with certain situations, usually when they encountered unpleasant, unmanageable situations or when telling would itself cause more pain, so they chose to avoid talking about it.

#### 3.1.2. Tendency to Avoid Emotional Language

Seven of the interviewees tended to avoid emotional language. This was manifested by not understanding emotions, understanding emotions with rationality, and replacing emotional expression with thoughts.

Amy and John made it clear that they did not know what their emotions were.


*John: But in fact, in the beginning, I didn’t feel that there was anything special about my emotions. Because I actually don’t know when I feel depressed.*


The statements by Amy, Bill, and Frank conveyed that they understood emotions rationally, rather than spontaneously feeling them and expressing them.


*Amy: I don’t know what emotions I’m feeling for myself. Because I sometimes may be in a certain situation, I think I should be happy, and I just act happy and tell myself I’m happy. Actually, I haven’t tried a lot of happy emotions, so I don’t have much emotional perception.*



*Bill: My wife didn’t understand that I said those words. I really felt ice in my body. I later learned that it’s definitely not going to work if it’s real ice.*


Bill’s statement describes his characteristics. He claimed that there was ice in his body but later realized that he felt cold in his body. Two respondents also reported that these thoughts became less frequent after taking medication.

Four respondents reported that they always had a lot of thoughts and ideas going on in the mind instead of changes in feelings. Two interviewees reported having fewer irritable thoughts after taking medicine.


*Daniel: I am thinking all the time in my head every day. It is the kind of cranky thinking that is not controlled by me. Unlike some people who like to think actively, I am the kind of person who wants to calm down and let go of my mind. But I can’t. There will always be something that bothers me. I can’t be without thinking.*



*Frank: I feel that the kind of strange thinking has become less. For example, if there is an interview at 4 p.m., I may keep it in mind and think about it all day before the interview. But after taking medicine, now I don’t think it is as frequent. It may just occasionally come to remind me that there is an interview in the afternoon.*


Daniel and Frank describe themselves thinking a lot in their heads, rather than feeling confused or feeling overwhelmed. It suggests that they avoid using subjective emotional language, and instead, express how frequently they were thinking.

In summary, not understanding emotions, rational understanding of emotions, and replacement of emotional expression with thoughts all suggest that patients with somatization symptoms tend to avoid emotional language. They do not understand their own feelings and emotions, and they are not good at expressing hot and cold or happy or sad. They sometimes understand through analysis that there are “happy” scenes and the existence of “stress”. They did not express that they were bothered by too many emotions, but by “thinking too much”. When others tried to confirm whether they were worried, they would deny it.

### 3.2. Semantic Level: Power and Contestation

At the semantic level, there is power and contestation between patients and doctors when explaining the symptoms. Patients sought medical treatment due to physical discomfort. During the process of seeing a doctor, they encountered a “shock” when the diagnosis and treatment results were beyond their cognition. However, as the treatment progressed, the patients accepted medical discourse and adjusted their interpretations, and some patients integrated and developed their interpretations. There are three sub-themes: a plain explanation related to the body, a plain explanation meets medical discourse, accepting medical discourse, and adjusting one’s own interpretation.

All 10 interviewees believed at the beginning that the discomfort was caused by somatic problems. They initially consulted the emergency department, as well as gastroenterology, neurology, and cardiology specialists, without considering clinical the psychology or psychiatry departments. However, the examination results could not explain their symptoms, or the treatment was not effective for a while, and then they came to the clinical psychology department for treatment. Among them, six interviewees accepted the doctor’s advice, two interviewees considered the history of mental illness in the family, and two interviewees learned that their symptoms may be related to mental illness through searching on the Internet. These 10 interviewees have all experienced the transition from the physical department to the psychological department and have been “shocked” to a certain extent.

#### 3.2.1. Plain Explanation Relates to the Body

Among the 10 interviewees interviewed for this study, no one had the same symptoms. There were slight differences in the description of symptoms, even for those at the same site. Cherry, Frank, and John all had symptoms in the stomach, but the manifestations were different. Cherry felt that she could not eat and vomited out everything she ate. Frank felt nauseated and wanted to vomit in some situations. John was belching and kept burping for most of the day. The symptoms of Daniel and Hebe were in the head. Daniel’s symptoms included dizziness and cervical pain, but Hebe presented with headache and various forms of discomfort in the head, including breath blowing, cool feeling, etc. Amy was suffering from low back pain. Bill was suffering from upper limb numbness. Emily was suffering from cervical pain. George was suffering from heart pain. Ivor was suffering from a feeling of being wrapped in the back and discomfort in his legs. Some patients showed a changing pattern of symptoms or had multiple sites of discomfort.

When each interviewee began to have physical symptoms, they would go to the corresponding department for diagnosis and treatment according to the parts of the body where the symptoms were experienced, including gastroenterology, cardiology, neurology, etc. Their choice of department for treatment was often related to their understanding of their symptoms. This was based on the patient’s simple understanding of their physical symptoms. Each patient talked about his or her initial understanding of symptoms.

Amy at first felt pain in the lower back and thought it was a back sprain so she went to see the sports injury department. Bill had various symptoms, such as chest pain, fear of sound, and numbness in the hands. Bill thought these symptoms were caused by surgery for clearing urinary stones and that his body was stimulated. Cherry felt that there was something wrong with her stomach because she could not eat and vomited out everything she ate. She also thought it was caused by stomach discomfort and low blood sugar when she fainted on the bus. Daniel thought that his dizziness and fainting were caused by a problem with his cervical spine. Emily had symptoms of dizziness and cervical pain, which she believed to be neurasthenia caused by being too tired when she was young. Frank’s symptoms were nausea and vomiting, which he initially thought were a stomach problem. George initially thought that his chest pain was caused by taking medicine for his cough. Hebe’s symptoms were multiple forms of head discomfort. She thought it was caused by the increased prescription strength of her glasses. Ivor felt back pain and back wrapping, which he thought was caused by his body structure, with the bones protruding. John had a gastroscopy in January, which found polyps, and subsequently underwent resection. He was diagnosed with non-atrophic severe gastritis and took gastric medicine for 2 months. The symptoms of belching occurred in May, which John believed to be related to gastric acid reflux and related to his family history as two family members had suffered stomach cancer.

From the plain explanations of the above 10 respondents, we can see that when they had physical symptoms, they first thought there was something wrong with the part of the body where the symptoms appeared. For example, low back pain may be a sprain, and vomiting or belching was interpreted as a stomach illness. At the same time, they also observed their symptoms and revised their explanations based on their own experience. For example, when Cherry fainted on the bus, she thought it was caused by hypoglycemia in the stomach; Hebe reported that she felt sore in the neck after changing glasses and thought it was because she was wearing glasses that were too strong, which added to her explanation of diminution of vision; John considered belching as another manifestation of gastric disorders in the context of his gastric polyps and gastritis, and in connection with several gastric cancer patients in his family, as having some genetic cause.

The interviewees’ plain explanations were usually based on a combination of physical sensations, common sense, and a certain logic, which often motivated their treatment. There is practical significance.

#### 3.2.2. Plain Explanation Meets Medical Discourse

When patients see doctors and suggest plain explanations, it is often found that the results do not directly support the explanation of symptoms. Some patients do not find relief after a certain period of symptomatic treatment. This is when they have the opportunity to come to the psychology department. During the process of visiting the psychology department, patients often feel that their physical problems are denied and even fall into a state of confusion. The doctor’s words play an important role in this process. All 10 interviewees mentioned the doctor’s words. They talked about how the doctor explained the patient’s condition and how these words affected them.

Patients often have test results that show no organic problems of the body, and then the doctors tell them that the test results are fine. However, the patients feel many physical discomforts, which creates large conflicting feelings in the patients.

Seven interviewees mentioned that when the doctor told them there was “nothing wrong” in their body or it was suggested they see a psychiatrist, the interviewees felt angry, frustrated, unaccepting, doubtful, confused, and even overwhelmed.


*Ivor: Yes, it was over in 5 min. I waited for half an hour but came out in 5 min. He just looked at a film and said I was fine. He even did not ask me serious questions. He just skimmed the film and then he said nothing wrong. I think doctors need to at least listen to what patients say but not rush to deny others. Patients don’t pretend to be sick, OK? That’s why I don’t like these doctors.*


When Ivor went to see a doctor, the doctor told him there was “nothing wrong” according to his test results, and the doctor did not listen to Ivor’s story. Ivor was very angry and felt denied by the doctor. During the interview, Ivor avoided answering the question of how he viewed the doctor’s explanation, which may be related to the denial he felt during the treatment. Bill and Ivor had similar feelings. Bill had a bad experience traveling to multiple specialist clinics in many places. He explained that he “went away in high spirits and returned in low spirits”. Bill felt that his physical discomfort existed objectively, and he did not believe it was a psychological problem. Amy and Hebe fell into a strong conflict and were at a loss. Amy reported that she had done various tests but could not find the physical reason and felt her body made her angry. Hebe expressed that she could not understand and wanted to go back for another test but was afraid of the same result. George questioned whether the doctor had completed all the possible tests. Cherry and Frank were not as intensely angry and overwhelmed as the other five.

In contrast, one interviewee described how the doctor did not simply tell him there was “nothing wrong”, but patiently explained, gave him confidence, and helped him establish the treatment decision.


*George: I feel like I can’t get well anymore. The doctor said she had seen someone who could not walk by herself get well after treatment. She said I can come by myself and I should be confident. I felt supported by the doctor. Confidence is really important for treatment. If you treat the patient to give him confidence and understanding, he will accept your treatment suggestions. If you always think that there is nothing wrong with him, he may not believe you.*


George believes that it is important for doctors to give patients confidence and understanding. If the doctor always denies the patient’s views, the patient will not trust the doctor, and there will be no effective treatment without a therapeutic alliance. George communicated with the doctor about his concerns about the side effects of the medicine. The doctor’s explanation gave George a sense of certainty, affirmed his concerns, and objectively helped him analyze the pros and cons of taking medicine, which played an important role in his treatment decision.

The doctor George felt trusted was the psychiatrist in the clinical psychology department, while the doctors mentioned by the previous seven interviewees were physicians they met first. The previous seven respondents did not directly state their feelings about the psychiatrist’s language, but they showed trust in the psychiatrist. It could be seen that they gradually accepted psychiatrist’s explanations to varying degrees. Compared with psychiatrists, physicians are more likely to view the disease from a biomedical perspective, lacking a multi-perspective of biology, psychology, society, and culture, and do not consider the acceptance of these patients in communication.

Four interviewees described how the doctors’ explanation resulted in them not being understood by their family members and not being able to explain their condition to the people around them, with the result that they fell further into helplessness.

Amy felt that there was no reason to ask for leave from school. Neither the results of various medical tests nor the diagnosis of a somatoform disorder in the psychology department could be a reason. It seems that in such a medical context and social environment, Amy feels very helpless, misunderstood, and not accepted. When George’s and Hebe’s family members learned that the doctors stated that there was nothing wrong with their bodies, these family members could not understand their pain and were even judgmental. The families’ suggestion of “not feeling the body” did not work for the patients. Hebe felt that “*there is a gap between us in every minute of our life*” and “*he’s talking too lightly*”. The patients fell into a lonely situation. Similarly, Ivor felt rejected by his mother’s understanding and suggestions.

Three interviewees described the importance of the doctor’s words to them. Patients sometimes repeatedly thought about what the doctor had said.

When Hebe went to the department of neurology, the doctor did not give direct feedback when asked about the symptoms but “smiled without saying a word”. Hebe felt this meant that the doctor thought she needed to go to the clinical psychology department. Hebe did not get a sense of certainty and acceptance from the doctor but felt denied and oppressed. The doctor’s words, facial expressions, and gestures all indicate their attitudes, which are important for patients.

When John was undergoing gastroscopy, the doctor asked him “Do you often have acid reflux?” John, therefore, speculated that the gastroscope showed that the esophagus had signs of damage. The doctor’s question during the examination deepened his speculation. John attached great importance to the doctor’s words. He took this sentence and thought about it over and over again, which affected his explanation and even treatment decision. John reported that he wanted to have another gastroscopy to find out the source of his belching symptoms, even though the doctor already told him there was no need to do it again in such a short time.

In the process of repeated communication with the doctor, Daniel accepted the doctor’s explanation that medication works on neurotransmitters. For Daniel, the new explanation also brought a new confusion. *Daniel: It seems to me that the substance only regulates negative emotions without positive ones. It’s very strange why the substance makes me uneasy, but not happy all the time.* Daniel seems to need more explanation from the doctor to resolve his confusion.

During the treatment process, the doctors’ words often have an important impact on the patient. When the doctor uses the test results to tell the patient that there is “nothing wrong“ in the body, it often results in denial and repression in the patient, and even affects the patient’s confidence in treatment so that they fall into confusion and despair. If the doctor can consider the patient’s feelings, establish a good doctor–patient relationship, and patiently explain, it can enhance the patient’s confidence in treatment.

The power of the doctors’ words becomes important when the patients’ plain explanations are confronted with medical tests. Patients feel conflicted between their own feelings and doctors’ explanations, which brings patients into a difficult situation.

#### 3.2.3. Accept Medical Discourse and Adjust One’s Own Interpretation

Patients felt angry and rejected that there was nothing wrong with their bodies. However, as their physical symptoms persisted, they had to come to the clinical psychology department. Some patients adjusted their attitudes towards psychological problems, some with a scientific research attitude, and others with a try-it attitude. The clinical psychology department was often their last choice. With the progress of treatment in the clinical psychology department, some of their symptoms were relieved, some changed form and some fluctuated. The patients also gradually adjusted their explanations, some abandoning plain explanations, some accepting the doctor’s explanations, and some integrating and developing their own explanations.
(1)Acceptance and development

Four of the interviewees accepted neurophysiological explanations.


*Daniel: As I have already had treatment for so many years, and the chief also talked with me about neurological substances in the morning, I think I agree with this point of view. I have also searched for relevant information. Indeed, when people are happy, it secretes something called dopamine, which will stimulate you to be happy.*


Daniel agreed with the doctor’s neurophysiological explanation and added his own experience when explaining. At the same time, Daniel also mentioned later that he found that his sensitivity decreased after drinking, and speculated that alcohol stimulated the nerves more slowly, so he was less frightened. Bill, Frank, and Hebe also reached similar understandings in communication with doctors or from knowledge online. They were able to apply these explanations to the changes they experienced after taking the medication.

All 10 interviewees received medication. The neurophysiological explanation is closely related to the mechanism of action of medication. Patients are more receptive to this explanation and more accepting of the doctor’s words when they obtain symptomatic improvement after taking the medicine. The biological explanation is more acceptable to the patients, which is closer to their plain explanation based on association with the body.

Five interviewees accepted psychological explanations and explored them.

In contrast to the initial anger and disbelief when hearing the doctor’s recommendation to see a psychiatrist, after a period of psychiatric treatment, all interviewees, except Ivor, were at least partially receptive to psychological explanations but showed varying degrees of depth in their exploration of psychological aspects.

Amy was an inpatient who underwent long-term psychotherapy to explore herself and to sort out her family relationships, and she gradually accepted the explanation of the impact of emotions on her body. Similar to Amy, Cherry, and Daniel were also transferred to the clinical psychology department after they had somatic symptoms. During the psychotherapy, they examined the influence of their original family, growth experience, and work pressure. They were able to use what they learned to adjust themselves and find a way to live.

Although Emily and John did not receive psychotherapy after they came to the clinical psychology department for treatment, they reflected on their life experiences and clearly expressed that they were under tremendous pressure. Emily described how she took care of her schizophrenic mother and adolescent son alone for years. John’s father had been diagnosed with stomach cancer a year before and he had been betrayed by friends. He also had age and marriage pressure.

The patient obtained a different understanding from the doctor compared with before and deepened his interpretation by processing and expanding this understanding in relation to his own experience.
(2)Compromise and confusion

In contrast to the five interviewees above who focused more on their own experiences and psychological processes, the other four did not undertake much psychological exploration. They simply looked back at their experiences after they had achieved some improvement and found possible psychological fluctuations or personality influences at some point. They added psychological factors into their explanations but did not want to improve their relationships or personal status through psychological exploration.


*Frank: I think the symptom was caused by psychological factors first, and then I didn’t take care of the symptom. After five or six years, the symptom may further affect my mental or physical status. I have not been able to reverse it completely through psychology methods, because it may involve some physical or neurological stuff. So I need to take some medicine to help me reverse it. But it cannot be denied that at the beginning, I was quite depressed during that time. I think I may have been depressed for a while, thinking life is meaningless every day.*


Frank, a graduate student in science, came to the clinical psychology department for treatment when his symptom continued to worsen for 6 years, saying he was like a researcher studying his illness. After obtaining some relief while on medication, he was more certain that he fit the diagnosis of somatoform disorder. He did not delve into his psychological state but rather used an analytical approach to examine the relationship between the mental and physical. Hebe, who expressed something similar to Frank, came to see a doctor because of a series of head symptoms. She also analyzed herself and admitted that she was anxious because of the increase in the strength of her prescription glasses when she first had symptoms. However, she was also very confused as to why she still had symptoms when she was currently more emotionally stable. The same confusion also appeared in Frank.

Most patients at least partially accepted the doctors’ explanations, and some agreed with or even developed them. However, acceptance was not permanent, and even with medication and psychological exploration, these patients only reported an improvement in symptoms, not their disappearance. Amy, who had been hospitalized three times in the clinical psychology department, still suffered from back pain. Cherry was also a regular patient in the gastroenterology department. Emily continued to feel uncomfortable and wanted to go to the neurology department for treatment. Hebe thought she had stopped working to eliminate stress and was confused about why she still had symptoms. Daniel, Frank, and George also reported that their symptoms were alleviated but they did not know when they would completely disappear.

The patients compromised some of their own explanations and incorporated psychological factors into their analysis and understanding, rather than accepting psychological explanations.
(3)Persistence and retention

Four interviewees expressed that they still wanted to find evidence of physical problems.

Hebe’s head pain was relieved after 5 months of treatment, but she continued to have symptoms such as “blowing”, “flowing water” and “soreness” and still felt uncomfortable. Hebe went to the neurology department again for a head CT scan. At the same time, she went to see the chief of the clinical psychology department. The chief told her she seemed not bad and asked her to have another CT scan. Hebe felt her view was accepted. Later, she went to the dentist because of toothache, and the dentist told her that all four wisdom teeth were decayed, which might be the cause of the headaches. After her wisdom teeth had been removed, her head felt much better.

Amy received medication and psychotherapy in the clinical psychology department along with treatment in rheumatology and folk remedies. She was hospitalized three times in the clinical psychology department and continued treatment for almost a year, after having been treated in various other departments for four years. She reported that even with all these treatments, the pain in her lower back did not decrease. The reason she chose to continue the treatment in the clinical psychology department was that she did have problems with her family, and she had been frustrated in her studies for the past year and hoped to receive help.

Cherry and Emily also chose to go to the gastroenterology and neurology departments for treatment due to the frequent occurrence of different somatic symptoms while still receiving treatment in the clinical psychology department.

When the doctors’ words cannot be fully accepted by patients, the patients can fall into a state of uncertainty. In order to resolve the residual symptoms, patients still turn to multiple departments for help. Sometimes, this does not affect their continued treatment in the clinical psychology department.

Three interviewees retained a genetic explanation.


*Hebe: I feel that the genetic factor seems to be quite strong, and the tie of blood is really hard to tell. There is a hint. I feel like something is wrong with me and it’s quite normal. I have always felt that I should have some problems in this area.*


Hebe’s grandmother committed suicide due to mental illness, and her father suffered from depression. When Hebe first had head symptoms, she quickly thought that it might be a mental problem and came to the psychiatrist. She thought there was a genetic effect.


*John: On the one hand, there may indeed be organic problems, because my family has inherited this condition of stomach problem. My father has stomach cancer now, and my grandpa had esophageal cancer. I also have two aunts who actually have stomach ulcers. It seems a heredity in my family.*


John considered that the physiological aspect of his condition had a genetic basis in the stomach.

Emily’s mother had schizophrenia. She compared her symptoms with her mother’s and was afraid they could be inherited.

The family member’s illness is something that is certain and is often something that is easier for the patient to hold on to. It is also an explanation that patients can retain.
(4)Symbolic explanations

Patients sometimes symbolically express their understanding.


*Bill: After taking 16 or 17 pairs of Chinese medicine, I felt like I was going to fall apart. It is said in traditional Chinese medicine that there was cold Qi in the body. I felt like there was ice and blood in my body. I could feel the process of exclusion of ice and blood. I also asked the doctor if it was excreted from the urine. After I took 6 pairs, I felt like I was going to fall apart. I wanted to sit like this, but I felt that the legs were not mine. I could not feel the nerves.*


In explaining the changes in his body, Bill explained that he started to “drain ice” after taking the medicine, which was related to the “cold Qi” in Chinese medicine on the one hand, and the “cold” he felt on the other. Bill also asked the doctor for information on neurotransmitters to further enrich his explanation. Bill’s explanation is a little absurd from the objective reality, but if seen from the perspective of the “Qi” of Chinese medicine or Bill’s level of feeling, it can be understood partly. The language he used has symbolic characteristics, different from actual “Qi” in the general sense. The description of “Draining ice” and “draining water” also has symbolic characteristics.


*Emily: I think it’s accumulated over time. After a long period of accumulation, it erupts suddenly, like a volcanic eruption. It also accumulates over time. In fact, you can’t see it coming out. I guess it is the way. I think so. I feel so. Because you may not be able to see it outside, except with special equipment. Maybe it has gradually accumulated inside. When it accumulated to this level, the outer layer of the shell couldn’t hold it anymore, it might suddenly erupt. This would be the more serious volcanic eruption problem. I think if I wanted to explain how it works, it might be possible this way.*


Emily explained that her physical symptoms accumulated day by day in the process of caring for her mother and showing tolerance towards her son, which became a volcanic eruption. Emily’s explanation includes her own imaginative description, incorporating her experience and feelings and forming a dynamic picture.

Bill and Emily have their own unique explanations. These two interviewees did not fully accept the doctor’s words and did not reject their own plain explanations. Instead, they combined their own experiences to process and integrate their own explanations. Although these did not fit the modern medical explanation, their descriptions showed their feelings and explanations for the changes in their bodies, which had a sense of the Chinese landscape with symbolic and imaginative characteristics.

In summary, the patient feels huge conflict resulting from surprise and anger when hearing the words “referral to the clinical psychology department” for the first time. The patient is forced into a power struggle with the doctor’s words. With the progress of treatment, emergence of curative effects, communication with doctors, and scientific knowledge of diseases, some patients give up plain explanations, accept the doctors’ words, and integrate their inner experience and medical explanations; some patients still feel confused even though they partly accept medical explanations; some patients still insist on finding evidence for plain explanations; and some rely on symbolic explanations.

## 4. Discussion

### 4.1. Theme 1: Utterance Level—Avoidance

Patients can show behavioral avoidance, such as not responding, avoiding certain topics, and avoiding certain people. It has been shown that individuals with somatoform disorders adapt to a culture of emotional avoidance characterized by difficulty communicating concerns and emotions related to conflict during social interactions with significant adults [20]. Patients are less able to identify and express stress-related cognitions, emotions, and feelings, and are less physically and emotionally self-exposed, which makes them vulnerable to stressors. Amy, Cherry, and Hebe in this study all stated in interviews that if they opened up to their families, they would not be understood and would feel more distressed, so they preferred to avoid talking to their families about their distress. In general, patients relieve stress through avoidance behaviors, which also prolongs their experience of stress.

Thus, the experience of stress persists silently, embedded as a bodily experience that is not expressed in words. Instead, patients learn to use specific kinds of inhibitory or distracting emotional or behavioral coping strategies rather than seeking social support, positive reconstruction, or problem-focused coping strategies [20]. When the interviewee Ivor in this study heard that his mother interpreted his symptoms as stress-related or advised him to relax, he had some reactions of disapproval, but simply responded “I don’t know” or “Maybe”. Ivor could not bear the confrontation with his mother and suppressed his attitude and thoughts. Such coping strategies caused the stress to have a deeper impact on the body.

The tendency in patients with somatization symptoms to avoid emotional language is consistent with previous findings. Previous studies have shown that patients with conversion disorder and functional somatic syndrome share the same deficits in emotional encoding and reporting when conveying the emotional content of stimuli in actions, which is consistent with previous findings in hospitalized patients with somatoform disorders. Difficulties in the transition from implicit to explicit processing of emotions, exacerbated by anxiety, may constitute a somatization mechanism [21].

The tendency in patients with somatization symptoms to avoid emotional language may be related to alexithymia. It has been shown that alexithymia is associated with somatization independent of physical illness, depression, anxiety, and sociodemographic variables [22]. “Difficulty in identifying sensations” in the factor scale of the TAS-20 scale is the largest commonality between alexithymia and somatization. A similar result was found in a study of patients with chronic pain. When patients were asked how they experienced emotions, they described states of tension and restlessness rather than distinct and separate emotions [23].

In addition, patients do not confine their emotions to the mind or the body but focus on external events or behaviors. They have difficulty relating their feelings to bodily sensations, activities, or fantasies. Somatic sensations associated with emotional arousal may be amplified and misinterpreted as symptoms of illness [24]. Amy talked about how her parents were usually overwhelmed when she had an emotional breakdown and left her alone to go out to work. Ivor’s mother mentioned that his father expressed belittlement of Ivor’s emotions. In such a family environment, the interviewees themselves could not handle their emotions and thus chose to avoid them. Previous research has identified a continuum of interrelated patterns of emotional avoidance as an adaptive bio-psychosocial process that develops in childhood and persists into adulthood, negatively impacting physical and emotional awareness, communicative coping skills, and physiological stress responses [20].

At the utterance level, avoidance encompasses behavioral avoidance and a tendency to avoid emotional language. The impact of the culture of avoidance on patients, the prolongation of stressful experience due to avoidance, and difficulties in recognizing and processing emotions have also been discussed.

### 4.2. Theme 2: Semantic Level—Power and Contestation

Patients with somatic symptoms first perceive physical discomfort and naturally consider problems related to the body so they come to the internal medicine department for diagnosis and treatment. Some of them can feel their emotional problems to some extent, but they do not make the connection between their physical discomfort and their emotions.

There is a tradition of discussing the relationship between the body and mind in Western philosophy, where the body is the opposite of the mind, emphasizing the mind–body dualism. All kinds of problems in the body should be purely material under normal circumstances, and the body and mind are two paths that are not directly related. The emotional state of an individual due to psychosocial problems can no longer be determined by the subjective moral, ethical, and religious rules of traditional society, but by objective and neutral universal scientific laws that determine whether the person is healthy or sick. The psychological solution is not conceived of as an ideological indoctrination but as a scientific treatment. Thus, the model of psychological problems and psychotherapy can enter the field of medicine. This is known as the phenomenon of psychologization in modern Western society. In contrast, the Chinese response to illness is bodily in nature, and the mind is only a functional definition. All ontological functions are based on the same body. Chinese traditionally do not distinguish illness from mental illness [25]. Especially when the patients’ symptoms are themselves physical, they will naturally explain the physical discomfort based on past experience.

Patients in this study with somatic symptoms interpreted their symptoms differently and in conflict with their physicians’ explanations, even feeling that the actual symptoms are denied, which is consistent with previous research results [5]. Patients with somatization symptoms yearn for existential identity, which affects their self-confidence, stress coping, symptom perception, and coping attitudes. Patients have difficulties with self-confidence and self-perception in terms of body perceptions, vulnerability, and needs, which negatively affects their attempts to gain recognition in social interactions [26].

It is concluded that the participants in this study do not know how to judge the competence of professionals. When confronted with conflicting information, they feel that they do not have enough knowledge to choose. Therefore, they are forced to choose to believe in the doctors [8]. In medical practice, doctors and patients are in an unequal position, and patients must turn to doctors to relieve discomfort. When there is a discrepancy between the doctor’s words and the patient’s habitual understanding, it intensifies the patient’s conflict. This is the power relationship discussed by Foucault, who argues that power is not just a form of oppression or repression but is also productive, producing acts or events [27]. In the doctor–patient relationship, the patient turns to the doctor and follows medical advice, which can also be described as a process of discipline. The patient listens to the doctor’s explanation which gradually becomes his or her own. It is also possible that the patient’s avoidance is associated with his or her language absence. When the doctor expresses a psychological cause in an explanation, the patient may need to directly confront their stress and emotional difficulties, and have the opportunity to review experiences instead of avoiding them. It is also an opportunity for the patient to unfold the experience to gain understanding and meaning. The therapeutic process itself is also an acquisition of mental health knowledge, which in turn, promotes the improvement of the patient’s attitudes and beliefs about mental illness [28].

It is suggested that there is a kind of cultural conflict. Chinese people’s physical discomfort is in a complicated position, as Chinese people are in a complex position to express physical discomfort on both emotional and somatic levels in their daily lives. Chinese traditional culture does not directly encourage the expression of physical and psychological pain, and people in pain should endure their condition, while others around them should observe and empathize. This is an important part of the traditional Chinese training in empathy or sympathy. Modern physicians, however, have learned the Western scientific language but have lost the ability to empathize in the traditional context. This language is unable to describe and convey the unexpressed pain of Chinese patients, resulting in communication barriers [13]. Layman’s beliefs about illness form a parallel but less-accepted system of explanations that reflect cultural, social, and political influences [5]. A study of men taking antidepressants in New Zealand found that taking antidepressants led to impairment of vitality and male-related abilities, with conflicts and tensions between cultural and social dimensions of autonomy and doctor’s guidance [9]. There are two parallel understandings, the medical and the secular. Additional interviews are needed for patients to reconcile the two understandings [5]. As interviewee George reported, the doctor explained patiently, responded to his doubts about the medication, gave him confidence in the treatment and helped him establish treatment decisions.

After the interviewees in this study accepted the medical explanation, they did not mention their plain explanations again, except for two interviewees who used symbolic explanations. One consequence of the healing experience can be the loss of one’s early functional ways of understanding, communicating, and coping with distress [8]. The patients with somatization symptoms first faced the shock of medical discourse in the process of seeing a doctor, gradually adjusting their own interpretations, accepting the doctor’s interpretations, and even developing them. Being in a dominant position in the power relationship, the patients’ inner understanding is often not respected. They have to learn medical discourse to understand their situation, but medical explanations sometimes put them in a predicament, and then they feel isolated or even confused. As with the relationship between patients and their families, patients are unable to confront the conflict and choose to avoid it. This power pairing is repeated in the doctor–patient relationship.

Two patients creatively explained their symptoms using imagery-based language. In traditional Chinese culture, the mechanisms that cause psychosomatic illnesses are assumed to be certain factors that are simultaneously related to the human spirit and life. These factors are further attributed to the imagery-based concepts of Qi, Huo, and Meridian. For patients who do not know much about medical theories, these concepts can be seen as metaphors for certain energies of life, which may be directly impacted by the outside world or as a result of external stimulation of the body. The ups and downs of the energy disturb the overall balance and normal state of the body system, resulting in various abnormal states. Through the analogy of natural phenomena such as water, wind, and fire, one can well understand how energy works and imagine these physiological processes occurring in one’s body. The treatment of such problems in traditional Chinese medicine also reflects the combined application of intuition and imagery-based concepts [25]. Bill received traditional Chinese medicine treatment for a period of time, and Emily received massage therapy, which may be related to their ability to explain their conditions in terms of imagery-based concepts.

At the semantic level, when Chinese patients experience somatic symptoms, they first believe that something is wrong with their bodies. They experience conflict and denial, and then gradually accept adjusting their explanations. We have examined the differences in mind–body relations between Chinese and Western groups, the issue of power between doctors and patients, the conflict between modern medical concepts and traditional culture, and the impact of Chinese medicine’s explanations of psychosomatic problems on patients’ explanations.

### 4.3. Suggestions for Managing Patients with Somatization Symptoms

It was found that patients with somatization symptoms avoided talking about particular thoughts or behaviors, and even viewed communication with others as a new source of stress. In fact, relationships are both a source of stress and a way to relieve it. It is possible for patients to reduce somatic symptoms if they are able to make those close to them aware of their stressors with the help of doctors during the treatment process. For example, Cherry and Hebe indicated they seemed more able to tolerate the pain of the symptoms when their family members understood their stresses. Therefore, it is suggested that helping people with somatization symptoms should be considered on an interpersonal level so that the patients’ suffering is understood by those close to them.

Chinese people attach great importance to family. For patients with somatization symptoms, it is recommended to use the strength of the family to help them. For example, Amy mentioned in the interview that her mother took her to various places for treatment. Hebe’s parents would help her massage the uncomfortable position. They felt that their family members’ care and attention were especially important to them. Amy even invited her parents for psychotherapy to explore the difficulties that existed in the family. Parents feel the influence of the family on their children and actively participate in the therapeutic process.

As mentioned in the previous discussion section, the relationship between mind and body in traditional Chinese culture is monistic rather than dualistic as in the West. When doctors meet patients with somatization symptoms, it is recommended that doctors take the patient’s acceptance into account when providing explanations. As seen in the results of this study, patients are able to accept neurophysiological explanations as well as psychological explanations when they become involved in treatment. Doctors should inform patients and explain conditions in an integrated and comprehensive manner in the process of diagnosis and treatment, instead of emphasizing the absence of physical problems or the existence of psychological problems in patients.

### 4.4. Strengths and Limitations

Stortenbeker reviewed the linguistic and interactional features of medical consultations about medically unexplained symptoms [29]. The analysis identified three dimensions including symptom recognition, double trouble potential, and negotiation and persuasion. This supports our findings on doctor–patient conflicts and reconciliation. However, while Stortenbeker focused on medical consultations, our study provides a qualitative exploration of patients’ lived experiences. Weigel examined psychotherapists’ perspectives on treating somatic symptom disorders, exploring the explanatory models used in psychotherapy [30]. The findings emphasized the importance of patient-centered explanatory models. This aligns with our recommendation for doctors to provide humanistic explanations attuned to patients’ acceptance levels. Other studies have quantitatively analyzed the language patterns of patients with somatic symptoms [31,32]. The linguistic perspective represents a relatively meticulous research angle.

Our study makes several pioneering contributions through its qualitative methodology and patient-centered focus. We utilized interpretative phenomenological analysis to foreground patients’ subjective lived experiences and explicate the meanings within their language. The philosophical perspective drawing on mind–body relations in Western philosophy provided novel contextualization. Expanding the cultural lens, we discussed Chinese patients’ mind–body conceptions. Examining China’s unique circumstances in which rapid modernization is colliding with traditional beliefs can inform more culturally-suitable treatment approaches. Our research contributes insights needed to improve doctor–patient relationships and potentially reduce healthcare costs. Overall, this study offers humanistic perspectives to advance research and practice for patients with somatization symptoms.

The interviewees selected for this study were limited to patients in the clinical psychology department, and the research results may not be extended to patients with somatization symptoms in other departments.

## 5. Conclusions

The research shows that patients with somatization symptoms are prone to adopt avoidant coping styles, often responding with avoidance in the face of possible conflict and tending to avoid emotional language in their interactions with others. Patients have their own plain explanations for their symptoms, which are usually related to their bodies. When they come to see psychiatrists, they often receive psychological explanations which make them feel frustrated and conflicted as they feel that the medical discourses do not correspond to their own feelings. Patients gradually accept the medical discourse and adjust their interpretations in the subsequent treatment process, and some adopt symbolic interpretations related to traditional Chinese culture. It is worth noting that doctors are authority figures for patients, and their words have a crucial influence on patients. Furthermore, there is a certain conflict between modern medical explanations and traditional Chinese cultural concepts of disease. Therefore, doctors need to pay attention to the cultural background of patients as well as their own interpretations during the treatment process. Doctors also need to be aware of the influence on patients of their own forms of verbal expression during the consultation process.

## Figures and Tables

**Table 1 healthcare-11-02811-t001:** Information of Participants.

Number	Gender	Age	Marital Status	Education Level	Symptoms	Diagnosis	Treatment
Amy	Female	19	Single	High School	Low back pain	Somatoform disorder	Medication + Psychotherapy
Bill	Male	59	Married	Undergraduate	Upper limb numbness	Somatoform disorder	Medication
Cherry	Female	30	Single	Master	Stomach pain	Somatic symptom disorder	Medication + Psychotherapy
Daniel	Male	31	Married	Junior college	Dizziness, cervical pain	Anxiety disorder	Medication + Psychotherapy
Emily	Female	43	Divorced	Junior college	Cervical pain	Somatoform disorder	Medication
Frank	Male	23	Single	Graduate student	Nausea	Somatic symptom disorder	Medication
George	Male	29	Single	Undergraduate	Heart pain	Anxiety disorder	Medication + Psychotherapy
Hebe	Female	30	Single	Undergraduate	Multiple forms of head discomfort	Somatoform disorder	Medication + Psychotherapy
Ivor	Male	18	Single	High school	Discomfort in the back and legs	Somatoform disorder	Medication + Psychotherapy
John	Male	37	Single	Undergraduate	Belching	Somatic symptom disorder	Medication

**Table 2 healthcare-11-02811-t002:** Identified themes, sub-themes and sub-topics.

Themes	Sub-Themes	Sub-Topics
Utterance level: avoidance	Behavioral avoidance	
	Tendency to avoid emotional language	
Semantic Level: power and contestation	Plain explanation relates to the body	
	Plain explanation meets medical discourse	
	Accept medical discourse and adjust one’s own interpretation	Acceptance and development
		Compromise and confusion
		Persistence and retention
		Symbolic explanations

## Data Availability

Data is available from the corresponding author upon reasonable request.

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
