# Peer review of "Changes in Patient Discourse: A Qualitative Study Based on the Treatment Experience of Chinese Patients with Somatization Symptoms"

_healthcare, 2023, doi:10.3390/healthcare11212811_

Round 1

Reviewer 1 Report

This is an interesting qualitative study that confirms existing knowledge about somatization symptoms through patient interviews. I have the following additional and specific comments:

1.     Some of the 10 subjects had different medical diagnoses. I feel that the subjects with anxiety disorders in this study may differ from those with somatic symptom disorders in terms of psychodynamics. Please comment on whether the results of this study can be consolidated despite these differences.

2.     Present themes, subthemes, and subtopics in a table for clarity

3.     I don't understand why the subject's expression of having a lot of thoughts in their head instead of a change in feeling is labelled as lacking emotional language (Page 6, lines 255-262: Daniel & Frank's remarks). I suggest that you consider a different way of naming the topic (Lack of emotional language).

4.     If the results and discussion were organized around topic clusters, I think it would be clearer.

5.     Please highlight suggestions for managing patients with somatic symptom disorders from a cultural psychiatric perspective

I wish you all the best in your revision work and hope to see this manuscript as a published research article.

Sincerely, your reviewer

Author Response

Dear reviewer,

Thanks very much for your time to review this manuscript. We really appreciate all your comments and suggestions. We have considered these comments carefully and tried our best to address every one of them. Please find the detailed responses below and the corresponding revisions highlighted in the re-submitted files.

Comment 1: Some of the 10 subjects had different medical diagnoses. I feel that the subjects with anxiety disorders in this study may differ from those with somatic symptom disorders in terms of psychodynamics. Please comment on whether the results of this study can be consolidated despite these differences.

Response: Thank you for your observation about the different medical diagnoses among the 10 subjects in our study sample. You raise an excellent point. In a psychodynamic perspective, the focus of attention for anxiety disorders and somatic symptom disorders is on the dynamical mechanisms behind them. However, our study does not examine the dynamical mechanisms behind the symptoms, but rather focuses on the subjective experience of the patients from a phenomenological perspective. All patients recruited for this study had somatic symptoms. In terms of subjective experience, both patients with anxiety disorders and patients with somatic symptom disorders experienced somatic discomfort physically and went through a similar process of seeking treatment. Therefore, even if the psychodynamics of the anxiety disorders and somatic symptom disorders are different, it is possible to be consolidated in this study.

Comment 2: Present themes, subthemes, and subtopics in a table for clarity.

Response: Thank you for your suggestion to present the themes, subthemes, and subtopics in a table. We have added this table to the revised version so that it is clear to present the findings.

Themes

Sub-themes

Sub-topics

Utterance level: avoidance

Behavioral avoidance

Tendency to avoid emotional language

Semantic Level: power and contestation

The plain explanation relates to the body

Plain explanation meets medical discourse

Accept medical discourse and adjust one's own interpretation

Acceptance and development

Compromise and confusion

Persistence and retention

Symbolic explanations

Comment 3: I don't understand why the subject's expression of having a lot of thoughts in their head instead of a change in feeling is labelled as lacking emotional language (Page 6, lines 255-262: Daniel & Frank's remarks). I suggest that you consider a different way of naming the topic (Lack of emotional language).

Response: Thank you for your thoughtful feedback on this sub-theme. We agree that “lack of emotional language” is not an appropriate name. It has been changed to “tendency to avoid emotional language” in the revised manuscript.

We have also added the following explanation in the revised version: “Daniel and Frank describe them thinking a lot in their heads, rather than feeling confused or feeling overwhelmed. It suggests that they avoid using subjective emotional language, instead expressing how frequent they were thinking.”

Comment 4: If the results and discussion were organized around topic clusters, I think it would be clearer.

Response: Thank you for the useful suggestion. We have revamped the discussion section of the article to four parts: 4.1 Theme 1: Utterance level: avoidance; 4.2. Theme 2: Semantic Level: Power and Contestation; 4.3. Suggestions for managing patients with somatization symptoms; 4.4. Strengths and Limitations

We have added “At the utterance level, avoidance encompasses behavioral avoidance and tendency to avoid emotional language. It has been discussed about the impact of the culture of avoidance on patients, the prolongation of stressful experience due to avoidance, and difficulties in recognizing and processing emotions.” to the discussion of section 4.1.

We have added “At the semantic level, when Chinese patients experience somatic symptoms, they first believe that something is wrong with their bodies. They experience the conflict and denial, and then gradually accept to adjust their explanations. It has been examined about the differences in mind-body relations between Chinese and Western groups, the issue of power between doctors and patients, the conflict between modern medical concepts and traditional culture, and the impact of Chinese medicine's explanations of psychosomatic problems on patients' explanations.” to the discussion of section 4.2.

Comment 5: Please highlight suggestions for managing patients with somatic symptom disorders from a cultural psychiatric perspective.

Response: Thank you for encouraging us to highlight suggestions for managing patients with somatic symptom disorders from a cultural psychiatric perspective.

We have added these suggestions to “4.3 Suggestions for managing patients with somatization symptoms”. The suggestions are as follows:

“It was found that patients with somatization symptoms avoided talking about particular thoughts or behaviors, and even viewed communication with others as a new source of stress. Actually, relationships are both a source of stress and a way to relieve it. It is possible for patients to reduce somatic symptoms if they are able to make those close to them aware of their stressors with the help of doctors during the treatment process. For example, Cherry and Hebe had indicated that it seemed to be more able to tolerate the pain of the symptoms when their family members understood their stresses. Therefore, it is suggested that helping people with somatization symptoms can be considered on an interpersonal level, so that the patients' suffering is understood by those close to them.

Chinese people attach great importance to family. For patients with somatization symptoms, it is recommended to use the strength of the family to help them. For example, Amy mentioned in the interview that her mother took her to various places for treatment. Hebe's parents would help her massage the uncomfortable position. They felt that their family members' care and attention were especially important to them. Amy even invited her parents for psychotherapy to explore the difficulties that existed in the family. Parents feel the influence of the family on their children and actively participate in the therapeutic process.

As mentioned in the previous discussion section, the relationship between mind and body in traditional Chinese culture is monistic rather than dualistic as in the West. When doctors meet patients with somatization symptoms, it is recommended that doctors take the patient's acceptance into account when make explanations. As seen in the results of this study, patients are able to accept neurophysiological explanations as well as psychological explanations when they get involved in treatment. Doctors could inform and explain in an integrated and comprehensive manner in the process of diagnosis and treatment, instead of emphasizing too much on the absence of physical problems or the existence of psychological problems in patients.”

Hopefully, we have made an adequate revision based on the comments. If there is still something else we need to do with the revision on our manuscript, please let me know.

Yours sincerely,

Wenting Shu

Reviewer 2 Report

Dear Authors

First of all, I would like to congratulate for doing an extensive research. The manuscript was written in detail explaining all the factors. Some minor queries and corrections from my side:

- Line 72 - correct the spelling of "chines".

- You recruited the participants from the clinical psychology department from March to August 2021.

Did they come for the first time visit to the clinical psychology department. Or did they come for follow-up visit. Give details in the methodology. If they came for the first time, mention in the inclusion criteria. If they came for follow-up, will it nor affect your results?

- Line 133 - The second interview was conducted in a timely manner.

On an average, after how many days, was the second interview conducted?

- Line 390 - The doctor George felt trusted was the psychiatrist in the clinical psychology department, while the doctors mentioned by the previous seven interviewees were physicians they met first.

So what did the previous 7 interviewees felt about the psychiatrist in the clinical psychology department? Did all 7 trust the psychiatrist?

- Line 401 - A feels very helpless, misunderstood, and not accepted,..

Is this "A" refer to Amy?

- Line 436 - The power of the doctors’ words comes to be important when patients’ plain explanations meet medical tests. 

Consider revising the words "comes to be important".

Author Response

Dear reviewer,

Thanks very much for your time to review this manuscript. We really appreciate all your comments and suggestions. We have considered these comments carefully and tried our best to address every one of them. Please find the detailed responses below and the corresponding revisions highlighted in the re-submitted files.

Comment 1: Line 72 - correct the spelling of "chines".

Response: Thank you for catching the misspelling of "chinese" on line 72. I have corrected it to the proper spelling in the revised manuscript.

Comment 2: You recruited the participants from the clinical psychology department from March to August 2021. Did they come for the first time visit to the clinical psychology department. Or did they come for follow-up visit. Give details in the methodology. If they came for the first time, mention in the inclusion criteria. If they came for follow-up, will it nor affect your results?

Response: Thank you for raising this important point about the participant recruitment. We did not differentiate whether the recruited participants were first-time visitors or not. Because our study asked patients to talk about their experiences of the treatment process including this clinical psychology department visit. We extracted data from the verbal expression of their experience of the process, so whether they were first-time visitors did not affect the data results. To make it clearer, I have added "No distinction was made as to whether participants were first-time visitors" to the revised manuscript.

Comment 3: - Line 133 - The second interview was conducted in a timely manner.

On an average, after how many days, was the second interview conducted?

Response: Thank you for pointing out that I did not provide the specific timeline for the second interview. To address your question, the second interview was conducted an average of 15 days after the initial interview, with a range of 10-21 days.

I have updated the manuscript to include these details: "The second interview was conducted in a timely manner, an average of 15 days after the initial interview (range 10-21 days)."

Comment 4: - Line 390 - The doctor George felt trusted was the psychiatrist in the clinical psychology department, while the doctors mentioned by the previous seven interviewees were physicians they met first.

So what did the previous 7 interviewees felt about the psychiatrist in the clinical psychology department? Did all 7 trust the psychiatrist?

Response: Thank you for raising this important point. This is an excellent question.

The previous 7 interviewees did not directly express whether they trusted psychiatrists verbally during the interviews, but it was shown in the results section 3.2.3 that they trusted in psychiatrists. They could accept the neurophysiological explanations of psychiatrists and part of the psychological explanations. For the sake of clearer expression, I have also added the following statement in the revised manuscript "The previous seven respondents did not directly state their feelings about the psychiatrist's language, but they showed trust in the psychiatrist. It could be seen that they gradually accept psychiatrist's explanations in varying degrees."

Comment 5: - Line 401 - A feels very helpless, misunderstood, and not accepted,..

Is this "A" refer to Amy?

Response: Thank you for raising this point. You are correct that "A" refers to Amy, one of the participants. To avoid confusion, I have updated the sentence in the revised manuscript to: "Amy feels very helpless, misunderstood, and not accepted."

Comment 6: - Line 436 - The power of the doctors’ words comes to be important when patients’ plain explanations meet medical tests.

Consider revising the words "comes to be important".

Response: Thank you for your valuable comment. In the revised manuscript, I have rephrased that sentence to: "The power of the doctors' words becomes important when patients' plain explanations meet medical tests." I believe this revised wording more clearly conveys the meaning that the doctors' words take on greater significance when medical tests appear to contradict patients' symptom descriptions.

Hopefully, we have made an adequate revision based on the comments. If there is still something else we need to do with the revision on our manuscript, please let me know.

Yours sincerely

Wenting Shu

Reviewer 3 Report

The article delves into the treatment experiences of Chinese patients presenting somatization symptoms. The authors conduct a thorough qualitative study, shedding light on the discourse patterns and changes in these patients throughout their treatment.

The study is comprehensive and offers valuable insights into the experiences of patients with somatization symptoms within a specific cultural context. The authors employ a qualitative approach, allowing for a deeper understanding of patients' perspectives and narratives. The research provides a strong foundation for comprehending the complexities involved in treating somatization symptoms, particularly within the Chinese population.

However, it is crucial for the authors to consider expanding their discussion to incorporate recent research developments in the field. Integrating more recent studies could enhance the article's relevancy and ensure that the readers are presented with the most current and comprehensive information. Additionally, highlighting the pioneering aspects of their research, such as unique methodologies or novel findings, would further distinguish their work and emphasize its significance within the broader academic discourse.

In conclusion, the article offers an insightful exploration of the treatment experiences of Chinese patients with somatization symptoms. While the authors have made commendable efforts to unravel the nuances of patient discourse, considering recent research and emphasizing the pioneering aspects of their work would enhance the article's overall impact and contribute to the ongoing discourse in this critical area of research.

Author Response

Dear reviewer,

Thank you for your thoughtful feedback and praise for our qualitative study exploring the treatment experiences of Chinese patients presenting with somatization symptoms. We appreciate you suggest us to expand the discussion to incorporate recent research developments in the field and highlight the pioneering aspects of our research. We have added this part in discussion as section 4.4 Strengths and Limitation:

Stortenbeker reviewed the linguistic and interactional features of medical consultations about medically unexplained symptoms[29]. The analysis identified three dimensions including symptom recognition, double trouble potential and negotiation and persuasion. This supports our findings on doctor-patient conflicts and reconciliation. However, while Stortenbeker focused on medical consultations, our study provides a qualitative exploration of patients’ lived experiences. Weigel examined psychotherapists’ perspectives on treating somatic symptom disorders, exploring the explanatory models used in psychotherapy[30]. The findings emphasized the importance of patient-centered explanatory models. This aligns with our recommendation for doctors to provide humanistic explanations attuned to patients’ acceptance levels. Other studies have quantitatively analyzed the language patterns of patients with somatic symptoms[31,32]. The linguistic perspective represents a relatively meticulous research angle.

Our study makes several pioneering contributions through its qualitative methodology and patient-centered focus. We utilized interpretative phenomenological analysis to foreground patients’ subjective lived experiences and explicate the meanings within their language. The philosophical perspective drawing on mind-body relations in Western philosophy provided novel contextualization. Expanding the cultural lens, we discussed Chinese patients’ mind-body conceptions. Examining China’s unique circumstance of rapid modernization colliding with traditional beliefs can inform more culturally-suitable treatment approaches. Our research contributes insights needed to improve doctor-patient relationships and potentially reduce healthcare costs. Overall, this study offers humanistic perspectives to advance research and practice for patients with somatization symptom.

  1. Stortenbeker, I.; Stommel, W.; van Dulmen, S.; Lucassen, P.; Das, E.; Olde Hartman, T. Linguistic and interactional aspects that characterize consultations about medically unexplained symptoms: A systematic review. Journal of Psychosomatic Research 2020, 132, 109994.
  2. Weigel, A.; Maehder, K.; Witt, M.; Löwe, B. Psychotherapists' perspective on the treatment of patients with somatic symptom disorders. Journal of Psychosomatic Research 2020, 138, 110228.
  3. Stortenbeker, I.; olde Hartman, T.; Kwerreveld, A.; Stommel, W.; van Dulmen, S.; Das, E. Unexplained versus explained symptoms: The difference is not in patients' language use. A quantitative analysis of linguistic markers. Journal of Psychosomatic Research 2022, 152, 110667.
  4. Al Salman, A.; Kim, A.; Mercado, A.; Ring, D.; Doornberg, J.; Fatehi, A.; Crijns, T.J. Are patient linguistic tones associated with mental health and perceived clinician empathy? JBJS 2021, 103, 2181-2189.

Hopefully, we have made an adequate revision based on the comments. If there is still something else we need to do with the revision on our manuscript, please let me know.

Yours sincerely

Wenting Shu

Round 2

Reviewer 1 Report

Thanks the authors for their acceptance of my comments and well addition revised. Best wishes for a good paper to be published.